# Investigation of Acoustic Properties of Poroelastic Asphalt Mixtures in Laboratory and Field Conditions

**DOI:** 10.3390/ma14102649

**Published:** 2021-05-18

**Authors:** Wladyslaw Gardziejczyk, Piotr Jaskula, Jerzy A. Ejsmont, Marek Motylewicz, Marcin Stienss, Piotr Mioduszewski, Pawel Gierasimiuk, Maciej Zawadzki

**Affiliations:** 1Division of Road Engineering, Faculty of Civil Engineering and Environmental Sciences, Bialystok University of Technology, 45E Wiejska St., 15-351 Bialystok, Poland; m.motylewicz@pb.edu.pl (M.M.); p.gierasimiuk@pb.edu.pl (P.G.); 2Department of Highway and Transportation Engineering, Faculty of Civil and Environmental Engineering, Gdansk University of Technology, 11/12 Gabriela Narutowicza St., 80-233 Gdansk, Poland; pjask@pg.edu.pl (P.J.); marcin.stienss@pg.edu.pl (M.S.); 3Automotive and Military Technology Division, Faculty of Mechanical Engineering and Ship Technology, Gdansk University of Technology, 11/12 Gabriela Narutowicza St., 80-233 Gdansk, Poland; jejsmont@pg.edu.pl (J.A.E.); pmiodusz@pg.edu.pl (P.M.); 4Laboratory Department, MTM SA Road Construction Company, 35 Hutnicza St., 81-061 Gdynia, Poland; mzawadzki@mtm-sa.com.pl

**Keywords:** poroelastic asphalt mixture, air void content, sound absorption coefficient, water drainability, water permeability, tyre/road noise, highly-polymer modified bitumen

## Abstract

Measures for the improvement of acoustic conditions in the vicinity of roads include the construction of pavement structures with low-noise surfaces with optimal macrotexture and the highest possible sound absorption coefficient. Laboratory evaluation of acoustic properties of a designed asphalt mixture before its placement in the pavement is a good solution. Currently, the most popular method for the determination of the sound absorption coefficient of various construction materials under laboratory conditions is the Kundt’s tube test. Sound absorption coefficient can also be assessed based on field and laboratory measurements performed using a Spectronics ACUPAVE System. Other parameters characterising the acoustic properties of road pavement courses include air void content and water drainability or permeability. The article presents an analysis of results of sound absorption coefficient obtained using a Spectronics ACUPAVE System and water drainability and permeability of poroelastic mixtures obtained both in laboratory and on test sections, in relation to air void content and grading of the mixtures. It was established that poroelastic mixtures containing an aggregate of maximum particle size of 5 mm are characterised by better acoustic properties than mixtures with a maximum aggregate particle size of 8 mm. Changes of crumb rubber aggregate grading and bitumen type (within the tested range of values) as well as the addition of lime have shown no evident influence on the sound absorption coefficient. Noise level values at the speed of 30 km/h according to the CPX method were measured as well. Relationships between sound absorption coefficient, water drainability/permeability, and air void content were determined. The performed analyses confirmed that Spectronics ACUPAVE System may be applied for evaluation of acoustic properties of asphalt mixtures in laboratory conditions, but further research is needed to reduce the uncertainty of the results.

## 1. Introduction

Improvement of the acoustic climate in the vicinity of roads can be achieved by constructing pavements with reduced noise levels. The acoustic efficiency of wearing course is associated with the technology of its production, macrotexture, and absorption of sounds that are generated at the tyre/road contact area. The influence of these parameters has been comprehensively presented in [1].

Air void content, thickness and arrangement of porous layers reduce air vibrations in tyre tread grooves, at the same time minimising the impact of aerodynamic phenomena on the emitted sound level. The sound absorption coefficient is a parameter that characterizes the acoustic properties of porous pavements. Apart from the thickness and porosity of the wearing course, tortuosity and resistivity of the air also have an impact on its value. Pratico et al. [2,3] presented an analysis of the manner in which tortuosity and resistivity can be determined on the basis of parameters characterizing the tested mixtures. Theoretical considerations on the calculation of these parameters, including the microstructure of pores, grain shape and porosity, are presented in the research by Attenborough and Howorth [4]. Maintaining good acoustic properties during the use of porous surfaces is related to clogging of the pores in the wearing course. This was initially confirmed on porous asphalt pavements (PA8 and PA11) by Alber et al. [5].

Macrotexture and service life also have an impact on the acoustic properties of low-noise road surfaces [6,7]. Miljkovic and Radenberg presented an interesting method of assessment of the influence of macrotexture on tyre/road noise by establishing such parameters as: mean profile depth (MPD), maximum amplitude of the wavelength spectrum, wavelength corresponding to the maximum amplitude, shape factor of the surface and shape length determined from the average shape factor, and wavelength corresponding to the maximum amplitude [8].

For standard layers of asphalt mixtures (stone mastic asphalt—SMA, asphalt concrete—AC), use of aggregate with the maximum grain size of 8 mm reduces the maximum rolling noise level by 2–4 dB compared to mixtures with the maximum aggregate grain size above 10 mm [6,7,9]. Noise level reduction by as much as 6 dB on new surfaces can be achieved by constructing a wearing course of porous asphalt (single and double layers) or in the form of very thin asphalt concrete (VTAC) and ultra-thin asphalt concrete (UTAC) [9,10,11].

Vaitkus et al. [12] pay attention to the balance between mechanical and acoustic durability when designing mixtures that reduce tyre/road noise. They have identified the most suitable aggregates and asphalts taking into account the climatic conditions.

Prospective solutions in the field of noise reduction include poroelastic road surfaces, in which rubber granulate is an important component [13,14]. It accounts for at least 20% of the mass of the asphalt–rubber–mineral aggregate mixture. The addition of rubber enables reduction of the stiffness of the pavement, which results in reduction of the tyre tread impact noise generation [15]. Apart from a significant addition of rubber granulate, poroelastic road surfaces may contain 20–40% air voids. Low stiffness modulus and high porosity can effectively mitigate noise, achieving results that can potentially reach differentials of up to 10 dB(A) [15,16]. Thus far, a broad research of poroelastic mixtures has been performed under the PERSUADE project (PoroElastic Road Surfaces for the Advanced Defense of the Environment) [16,17,18,19]. The SEPOR project (Safe Eco-Friendly Poroelastic Road Surface)—which is being realised by the consortium of the Gdansk University of Technology, the Bialystok University of Technology, and the MTM SA construction company from Gdynia—is to some extent a continuation of the PERSUADE project, since its primary objectives include reduction of pavement noise without simultaneous loss of the heightened durability [20].

Noise associated with road pavement type is evaluated based on measurements of the maximum sound levels using the CPX, SPB, CPB and OBSI methods [6,21,22,23]. Mioduszewski and Gardziejczyk presented an assessment of inhomogeneity of low-noise wearing courses using the close-proximity method [24]. The CPX method enables evaluation of acoustic performance of new pavements and monitoring of their acoustic performance in real traffic conditions over their lifetime [25]. In the studies of Tonin et al., the differences between tyre/road noise according to the OBSI and CPX methods were established [26]. Cesborn and Klein, based on tests on pavements in France and Germany, determined the differences between test results obtained using the CPX and CPB methods [27].

Laboratory evaluation of acoustic properties of a designed asphalt mixture before its placement is a good solution. Currently, the most popular method for determination of sound absorption coefficient of various construction materials is the Kundt’s tube test [28]. It is also possible to determine the value of sound absorption coefficient using the Spectronics ACUPAVE System [29], which enables measurements both in field and laboratory conditions.

Sound absorption coefficient has been the subject of many studies. For example, the acoustic properties of asphalt concrete, SMA, porous asphalt and noise-reducing asphalt mixtures were compared [30]. The influence of asphalt mixture type, granulometry, and air void volume on sound absorption coefficient was investigated [31]. Chu et al. [32], apart from research on the influence of the above-mentioned parameters, investigated the influence of degree of clogging of pores on sound absorption.

It is difficult to achieve consistently repeatable results in research of the sound absorption coefficient using the two-microphone method [33]. This is due to the large number of variables, such as: sample cutting and preparation, sample fit and position in the tube, and sample material variability. An additional problem is that there is no reference sound absorbing material to which one could compare the results.

Possibility of assessment of acoustic properties of poroelastic mixtures with varying air void content and composition characteristics was verified within the scope of the SEPOR project. Changes in the composition of mixtures encompassed differences in the grading of mineral and rubber aggregate, addition of hydrated lime and modification of bitumen binder. In addition to measurements of the sound absorption coefficient, tests of water permeability of samples and drainability of pavement courses of poroelastic mixtures were performed as well.

The asphalt-rubber-mineral mixtures were designed at the Gdansk University of Technology, and the measurements of sound absorption coefficient using the Spectronics ACUPAVE System (using test slabs and cylindrical specimens) and water drainability/permeability tests (both in situ and laboratory methods) were performed at the Bialystok University of Technology. Based on the laboratory test results, a number of poroelastic mixtures was selected for placement on short test sections. Measurements of sound absorption coefficient, water drainability and noise level using the CPX method were performed on the test sections.

## 2. Materials

After initial field validation of SEPOR mixtures designed with crumb rubber aggregate and SBS-modified bitumen, as described in detail in [34,35], it was noted that further adjustment of the design was necessary, mainly due to the need to achieve higher air void content, better acoustic properties and higher internal cohesion of the asphalt-rubber-mineral mixture.

Coarse crushed gneiss aggregate, fine gneiss aggregate and limestone filler were used. The rubber part of the mixture was composed of crumb rubber obtained from tyre recycling.

In order to reduce the difference in stiffness between the binder and crumb rubber aggregate, highly SBS-modified bitumens 45/80-80 and 65/105-80 were also investigated. To improve the affinity of asphalt and aggregate, an addition of hydrated lime was applied and investigated [36]. Warm-mix additive Sasobit was also used in one variant to verify whether such an additive would improve workability and compaction of the poroelastic mixture. After the laboratory test phase at the Gdansk University of Technology, seven poroelastic mixtures were selected for the next stage of laboratory research at the Bialystok University of Technology. A summary of the variants that were used is presented in Table 1. Gradation curves of selected mixtures are presented in Figure 1a–c.

The amount of crumb rubber and bitumen was constant for all mixtures. The article does not give more detailed information on the composition of the asphalt-rubber-mineral mixtures, since it is intellectual property of the producer.

Each mixture was used to form 2 or 3 slabs with dimensions of 40 × 30 × 4 cm^3^, according to the EN 12697-33 [37] standard. An example test slab of mixture No. 3 is presented in Figure 2a, and cylindrical specimens cut out of this slab are presented in Figure 2b.

Air void content and internal shear strength was determined for each poroelastic mixture. No significant differences in air void content were noted between the mixtures No. 1 and No. 2, as well as among the four mixtures: No. 3, No. 5, No. 6, and No. 7. Significant differences were noted between the pair of mixtures No. 1/No. 2 and the remaining mixtures. Significant differences were also noted between the mixture No. 4 and the mixtures No. 6 and No. 7. A summary of the air void test results and internal shear strength is presented in Table 2.

## 3. Scheme and Scope of the Tests

The sound absorption coefficient and water drainability/permeability tests were performed on test slabs, as well as on cylindrical specimens cut from the slabs.

Sound absorption coefficient was investigated using the Spectronics ACUPAVE System (Lexington, KY, USA), which enables measurement of sound absorption coefficient of road surfaces in situ by means of spot measurement method, according to the requirements of the standard ISO 13472-2:2010 [29].

The ACUPAVE set comprises of the following main components: metal tube with handles, sound source—a JBL 2426K speaker (Northridge, CA, USA), PCB Piezotronics two 1/2” microphones (Depew, NY, USA) placed in holes in the wall of the tube, DT9837A data acquisition and processing module, ACUPAVE software v. 4.3, and a computer.

Figure 3a shows sound absorption coefficient measurement performed on a test slab. According to the ISO 13472-2:2010 [29] standard, three spots for sound absorption coefficient measurement were defined on each slab. In the case of considerable differences between the results, an additional fourth spot was defined on some slabs and the most extreme outlier among the four results was eliminated.

Cylindrical specimens were cut out afterwards from the test slab areas that had been subjected to sound absorption measurement (Figure 3b).

At the Bialystok University of Technology, the Spectronics ACUPAVE System has been adapted to sound absorption coefficient measurements on cylindrical specimens with diameter of 100 mm, based on the model of Kundt’s tube tests (Figure 4) and according to the requirements of the standard ISO 10534-2:1998 [38].

Water drainability of the slabs was tested using the in-situ method (according to the EN 12697-40:2020 standard [39]). In the case of the in-situ method (Figure 5), the time of outflow was measured for 4 litres of water.

Vertical water permeability was tested and assessed in laboratory conditions according to the EN 12697-19 +A1 standard [40]. The test consists in measurement of the rate of vertical water filtration through a specimen loaded with a column of water of constant height, at ambient temperature of 15 °C to 25 °C. Vertical permeability *K_v_* (m/s) is calculated according to the Darcy’s law:*K_v_* = (4 × *Q_v_* × *l*)/(*h* × *πD*^2^),(1)
where:-*Q_v_*—vertical flow of water through specimen (m^3^/s),-*l*—specimen thickness (m),-*D*—specimen diameter (m),-*h*—constant height of the water column (in vertical permeability test) (m).

Vertical water permeability of cylindrical specimens according to the so-called laboratory method was determined using the device shown in Figure 6. The obtained results were evaluated according to the provisions of the [40] standard, which states that water permeability of porous pavements should be within the range from 0.5 × 10^−3^ m/s to 3.5 × 10^−3^ m/s.

Tyre/road noise was measured on test sections by means of the CPX method according to the current version of the ISO 11819-2:2017 standard [22], using a specialised test trailer, namely Tiresonic Mk 4 (Figure 7) [41].

The near-field measurements were performed using two microphones installed in the close proximity of the tyre/pavement contact surface. Due to the arrangement and length of the test sections, the noise measurements were carried out only at the speed of 30 km/h. Two reference tyres were used according to the ISO 11819-3:2017 [42] standard: one representing the noise from passenger vehicles (P225/60R16 Uniroyal Tigerpaw “Standard Reference Test Tyre” (P1), Greenville, SA, USA) and one representing the noise from heavy vehicles (195R14C Avon Supervan AV4 (H1), Kansas City, MO, USA). Based on the obtained results, the CPX index (CPXI) was calculated, which is the arithmetic mean of noise levels measured for the two reference tyres (each tyre is assigned the weight of 50%).

## 4. Laboratory Test Results and Their Analysis

Figure 8 presents example results of sound absorption coefficient measurements on test slabs (s) and cylindrical specimens (c) of poroelastic mixtures No. 3 (two slabs: 3.1, 3.2; average value labelled as 3s/3c) and No. 5 (three slabs: 5.1, 5.2, 5.3; average value labelled as 5s/5c) at eight 1/3-octave frequency bands. The data shown confirm the high variability of the results. Other researchers have also pointed out this problem [3,33].

A comparison of average values of sound absorption coefficients obtained for test slabs (labelled as 1s–7s) is shown in Figure 9, and for cylindrical specimens (1c–7c) in Figure 10. In the case of slabs, the highest values were obtained for mixture No. 7 and—slightly lower—for mixtures No. 5 and No. 6. The results obtained among slabs of mixtures Nos. 1–4 were comparable, with the lowest values measured for the mixtures No. 1 and No. 2, which were characterised by the lowest air void content. Analysis of the sound absorption coefficient results obtained for cylindrical specimens shows that the differences in values are greater than for the test slabs, but the ranking of the mixtures remains the same.

Greater values of sound absorption coefficient obtained for the cylindrical specimens are related to different test conditions. During the test, cylindrical specimens are placed in an aluminium pipe extension connected to the base of the Spectronics ACUPAVE System set.

Slightly higher sound absorption coefficients were noted for slabs and specimens of mixtures containing mineral aggregate of maximum particle size of 5 mm (excluding mixture No. 4).

The results of the test do not reveal any evident influence of changes in crumb rubber aggregate grading, bitumen type or lime addition on the values of sound absorption coefficient. This is due to insufficient diversity of the analysed factors. Mixtures with aggregate grain size of 5 mm and 8 mm and rubber granules of 2/4 mm and 4/5.6 mm were tested. However, certain regularities should be noted. In the case of tested slabs, the maximum values of sound absorption coefficient (less than 0.40) were measured in the frequency range of 800–1000 Hz. Its values also increased at the frequency of 1600 Hz in the case of slabs with a maximum aggregate grain size of 8 mm (Nos. 1–3) and in the case of slabs of mixture No. 4, with a maximum aggregate grain size of 5 mm, but with the lowest voids content in this group of slabs.

A more pronounced influence of air void content was noted in tests on cylindrical samples. The maximum values of sound absorption coefficient (less than 0.60) occurred at the frequencies of 400–800 Hz.

A comparison of sound absorption coefficient results obtained for test slabs (1s–7s) and cylindrical specimens (1c–7c) shows differences between the measured values depending on the test method, as well as some similarities in terms of the ranking of the tested mixtures (Figure 11). In this case, the effect of air void content and the maximum grain size of the aggregate on the average values of sound absorption coefficient for the frequency range of 315–1600 Hz is also visible.

The influence of air voids on sound absorption coefficient for mixtures with maximum aggregate particle sizes of 8 mm and 5 mm is presented in Figure 12. Despite some differences noted between the results obtained on test slabs and on cylindrical specimens, a significant influence of air void content on absorption coefficient was confirmed both for the test slabs (Figure 12a) and the cylindrical samples (Figure 12b).

The relationships shown in Figure 12 confirm the influence of air void content on sound absorption coefficient. Mixes Nos. 4–7 (based on aggregate with a maximum grain size of 5 mm) are characterized by better acoustic properties than mixes Nos. 1–3. However, it was noted that the effect of an increase in air void content on sound absorption coefficient of the test slabs and cylindrical samples was slightly different. In the case of slabs, an increase in air void content by 5% in the PSMA8 mixture increased the sound absorption coefficient by 0.08, and in the PSMA5 mixture—by 0.12. In the case of cylindrical samples, the coefficient increased by 0.10 and 0.08, respectively.

The problem of the influence of air void content, layer thickness, and aggregate grading was analysed by other researchers. Vaitkus et al. [30] noted that the highest sound absorption (more than 0.40) for PA8 mixture occurred in the frequency range of 700–1200 Hz, while in the range of 800–950 Hz it was higher than 0.80. Other tested mixtures showed good results at the frequency range of 350–550 Hz.

Knabben et al. [31] noted small differences between the acoustic behaviour of rubberized porous courses with maximum aggregate size of 9.5 mm and 12.5 mm. They also confirmed that there is no significant difference in sound absorption between a typical dense-graded asphalt mixture and a dense-graded rubberized asphalt mixture.

Zhang et al. [43] found out that an excessive content of air voids does not significantly improve the acoustic properties of the mixtures. In their opinion the optimal solution is the content of voids in the range of 17–24%. They also indicate the influence of aggregates with grain size of 1.18 mm and 2.36 mm on sound absorption.

Chu et al. [32] showed that sound absorption of porous asphalt mixtures was the most considerable in the frequency range of 250–1250 Hz. They also found that the sound absorption capability of the mixtures within the frequency range of 250–1000 Hz was affected to the greatest degree by clogging of the mixture.

The obtained test results and the results presented in the references confirm that further research is needed on the influence of the parameters characterizing the mixtures on their acoustic properties.

Results of water drainability of the test slabs and vertical water permeability of the cylindrical specimens are presented in Figure 13. The time of water outflow in drainability tests of the slabs of mixture No. 1 was considerably longer than 180 s, so it was omitted in Figure 13a.

Water drainability/permeability analysis shows agreement between the results of the two methods. The best results were obtained for mixtures No. 3, No. 5, No. 6, and No. 7.

Figure 14 shows evident influence of water drainability/permeability (determined both as the outflow time using the in-situ method and as vertical permeability using the laboratory method) on sound absorption coefficient of poroelastic mixtures.

## 5. Field Verification of Laboratory Tests

After the phase of laboratory tests performed in Gdansk and Bialystok, the following mixtures were selected for use on short test sections: No. 1, No. 3, No. 4, and No. 5 (Figure 15).

For all variants, highly polymer-modified binder 45/80-80 was used. The mixtures were laid and compacted on an access road near an asphalt plant. Typical batch type asphalt plant was used. The construction process involved changes of the following technical factors to investigate their potential influence on the properties of the poroelastic layer:different types of asphalt paver—small and normal size;different levels of initial compaction by paver screed (using only tamper bars or tampers and vibrating plates);different levels of final compaction provided by an 8.5-ton roller (three or seven passes).

In total, 18 different short test sections were constructed. Their arrangement is presented in Figure 16. The analyses presented in this article encompassed pavements on test sections from No. 5 to No. 16 (Table 3). Sound absorption coefficient and water drainability of the mixtures were not measured on sections Nos. 1–4, No. 17, and No. 18.

Water drainability and sound absorption coefficient were measured on the constructed test sections. Their results are shown in Figure 17. The colours represent the types of poroelastic mixtures used on particular test sections (red—mixture No. 1, green—mixture No. 4, blue—mixture No. 5, yellow—mixture No. 3). Figure 18 presents the values of the CPX index measured at the speed of 30 km/h.

Based on the field test results, the effect of air void content on sound absorption coefficient was determined for mixtures containing aggregate with maximum particle sizes of 5 mm (black points) and 8 mm (grey points), as shown in Figure 19a. Relationship between water drainability obtained using in situ method and the sound absorption coefficient was also determined (Figure 19b). The sound absorption coefficient and water drainability were measured at the same locations on the surface of the test sections.

The results obtained from the test sections have confirmed that laboratory test results might be used as the basis for selection of mixture for the wearing courses on real road sections. Based on the results of tests performed in laboratory and on short test sections, a number of poroealstic mixtures were chosen for placement on long test sections within the next stage of the SEPOR project.

## 6. Conclusions

Based on the performed tests and analyses, the following conclusions may be formulated:Air void content significantly affects both the sound absorption coefficient and the water drainability/permeability of poroelastic mixtures.The use of a Spectronics ACUPAVE System enables ranking of poroelastic mixtures according to their sound absorption ability. However, the results obtained for cylindrical specimens are slightly higher than for test slabs.Both the use of in situ method and the laboratory method of water drainability/permeability measurement enables the evaluation of acoustic properties of poroelastic mixtures.Poroelastic mixtures containing aggregate of maximum particle size of 5 mm are characterised by better acoustic properties than corresponding mixtures with maximum aggregate particle size of 8 mm.Changes of crumb rubber aggregate grading and bitumen type (within the tested range of values), as well as the addition of lime, have shown no evident influence on the sound absorption coefficient.Noise level measurements on test sections using the CPX method confirmed the ranking of the used poroelastic mixtures in terms of their acoustic properties.The results of the Spectronics ACUPAVE System tests are subject to relatively high uncertainty. Hence, further research will be carried out to improve the reliability of this method.Research work will be continued in order to obtain more unambiguous findings regarding the influence of the material and technological characteristics of poroelastic asphalt mixtures on the acoustic properties of road surfaces.

## Figures and Tables

**Figure 1 materials-14-02649-f001:**
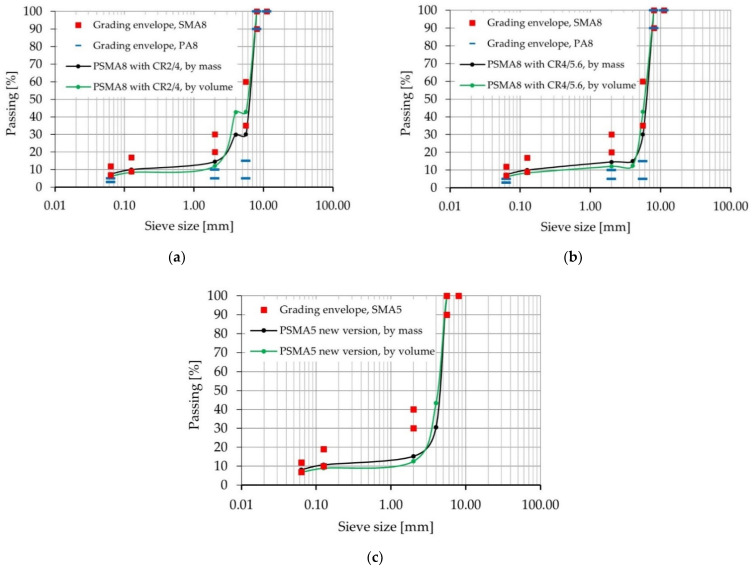
Gradation curves: (**a**) PSMA 8 CR 2/4, (**b**) PSMA 8 CR 4/5.6, (**c**) PSMA 5 CR 2/4.

**Figure 2 materials-14-02649-f002:**
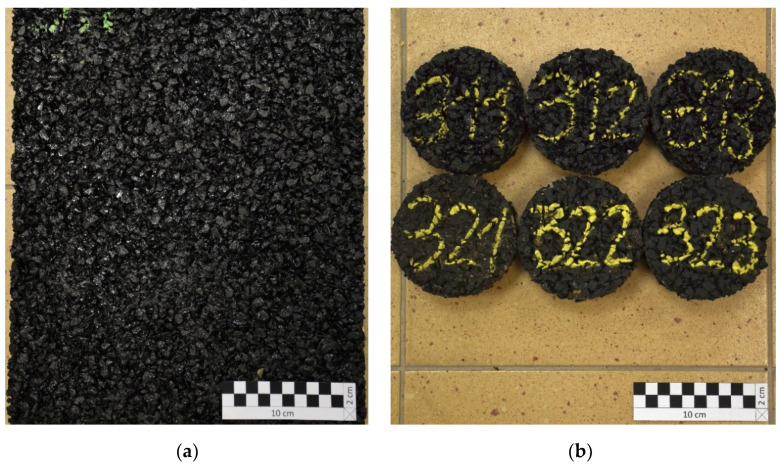
Mixture No. 3—a slab (**a**) and specimens (**b**).

**Figure 3 materials-14-02649-f003:**
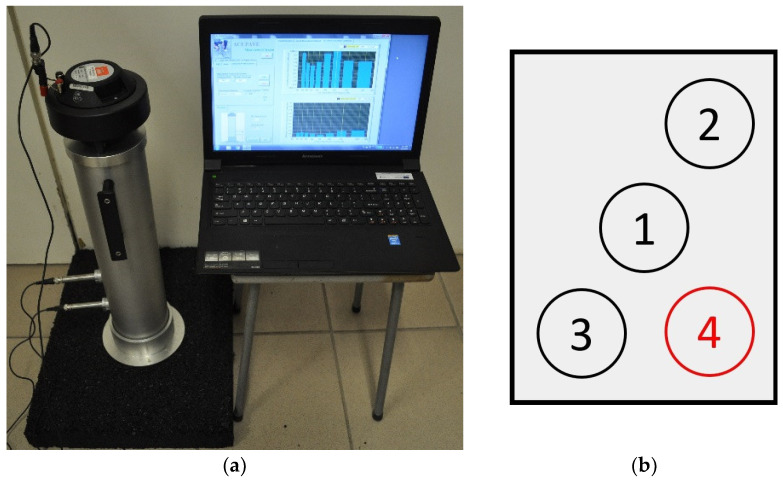
Sound absorption coefficient measurement on a test slab (**a**) and locations of measurements and coring of cylindrical specimens (**b**).

**Figure 4 materials-14-02649-f004:**
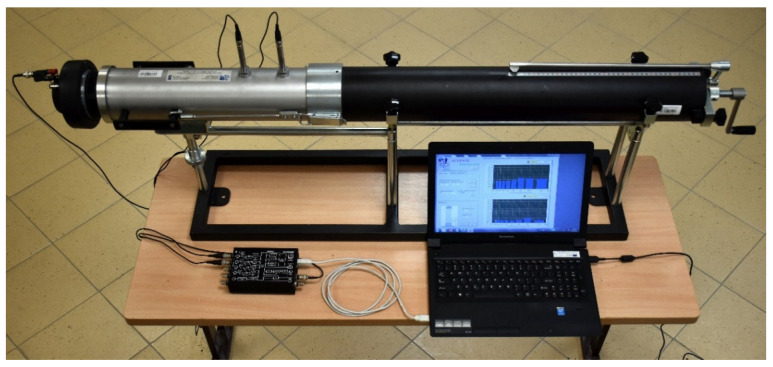
Spectronics ACUPAVE System measurement set.

**Figure 5 materials-14-02649-f005:**
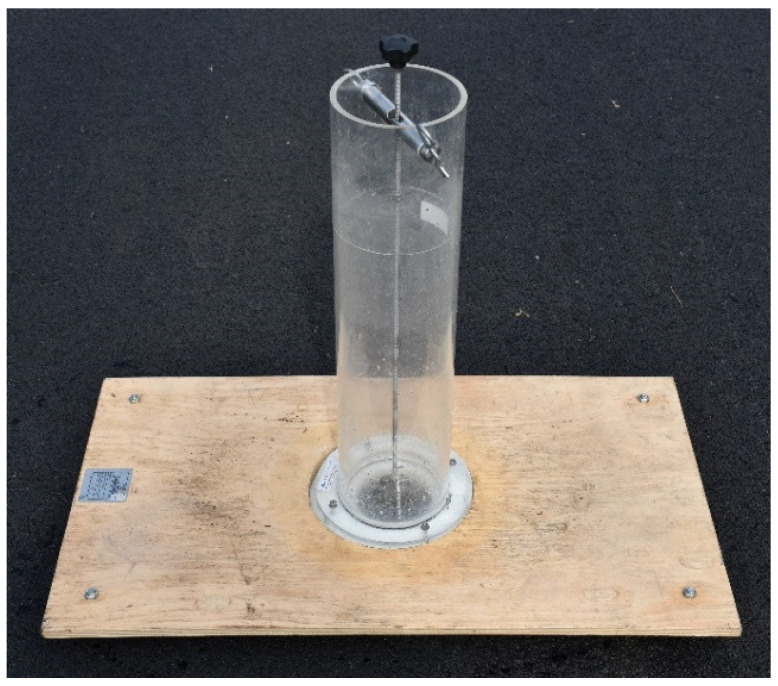
Water drainability testing device for the in situ method.

**Figure 6 materials-14-02649-f006:**
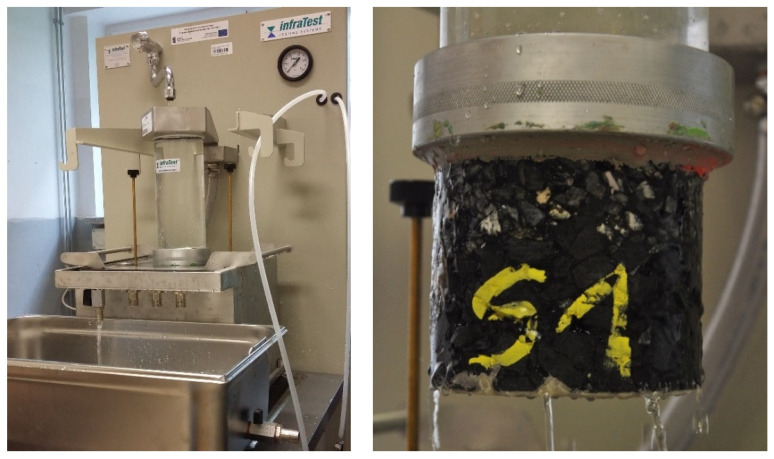
Vertical water permeability test.

**Figure 7 materials-14-02649-f007:**
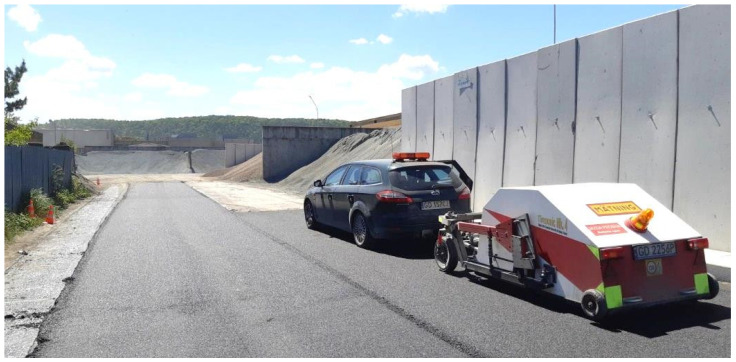
Tyre/road noise measurements using the CPX method.

**Figure 8 materials-14-02649-f008:**
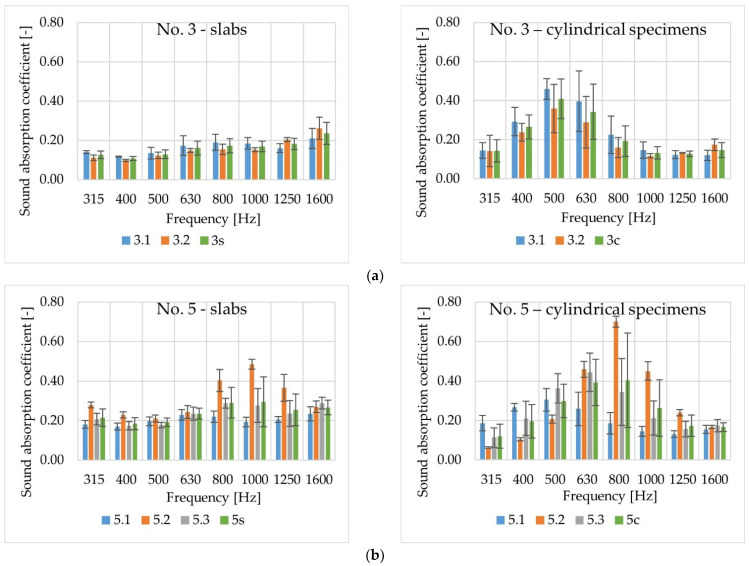
Sound absorption coefficients of poroelastic mixtures No. 3 (**a**) and No. 5 (**b**).

**Figure 9 materials-14-02649-f009:**
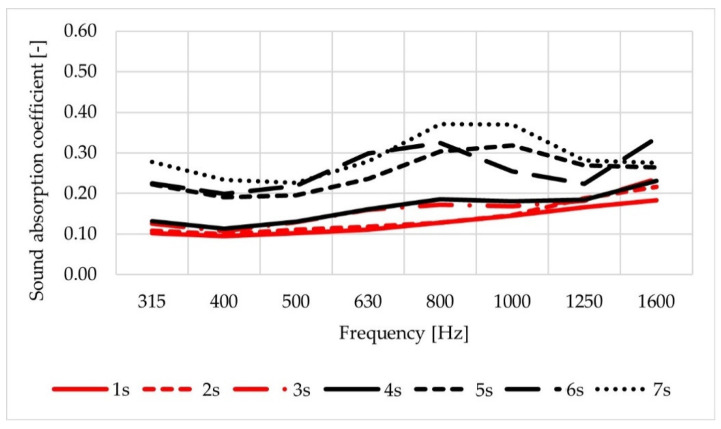
Average values of sound absorption coefficient obtained for the test slabs.

**Figure 10 materials-14-02649-f010:**
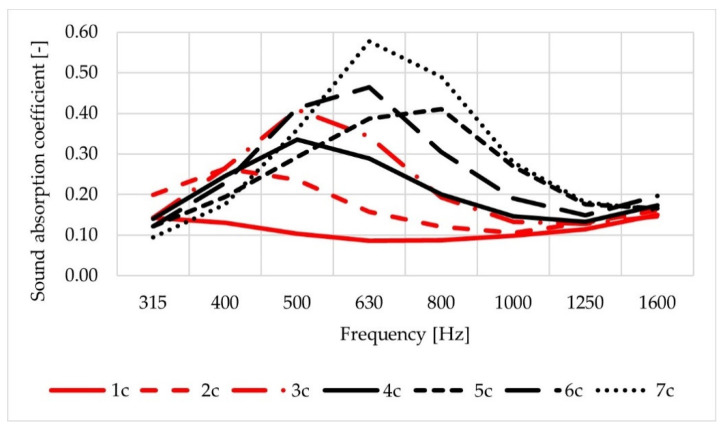
Average values of sound absorption coefficient obtained for the cylindrical specimens.

**Figure 11 materials-14-02649-f011:**
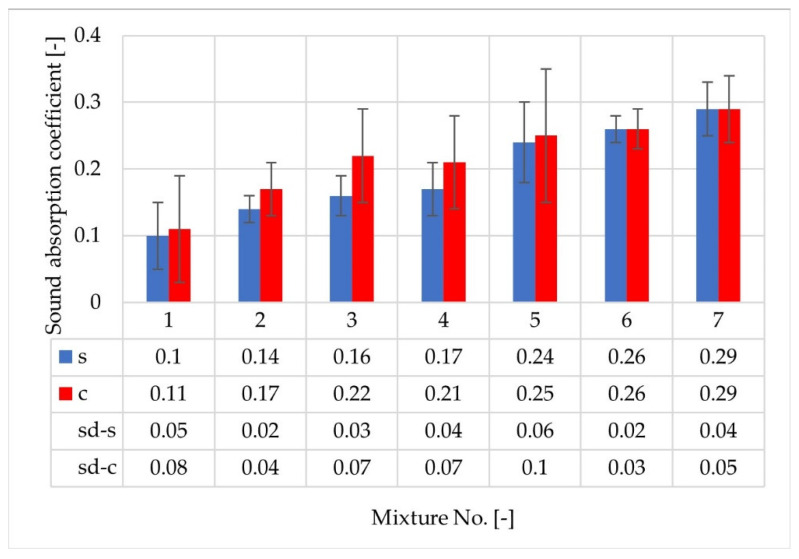
A comparison of the average sound absorption coefficient values obtained for test slabs (s) and cylindrical specimens (c).

**Figure 12 materials-14-02649-f012:**
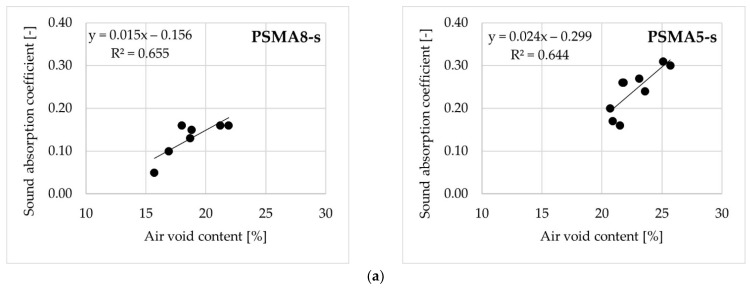
The influence of air voids on sound absorption coefficient: test slabs (**a**) and cylindrical specimens (**b**).

**Figure 13 materials-14-02649-f013:**
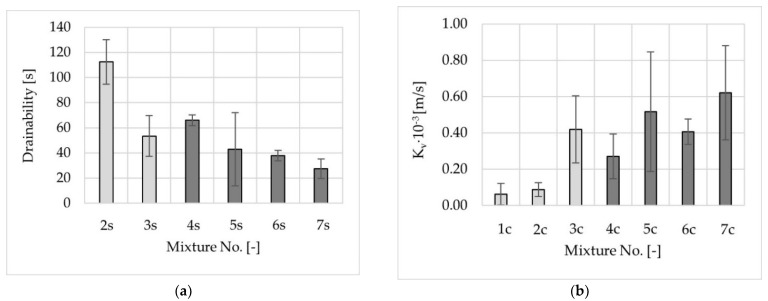
Water drainability of test slabs (in situ method outflow time) (**a**) and vertical water permeability of cylindrical specimens (**b**) of poroelastic mixtures.

**Figure 14 materials-14-02649-f014:**
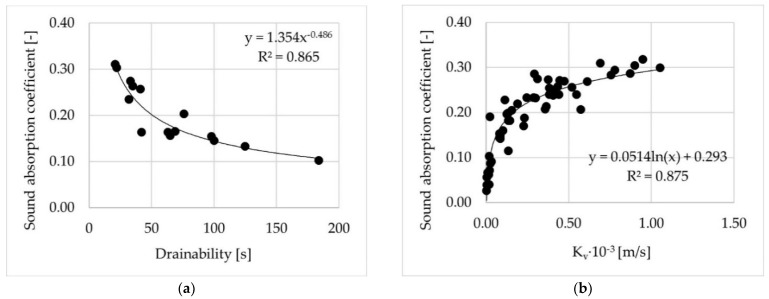
Relationships between water drainability/permeability and sound absorption coefficient for test slabs (**a**) and cylindrical specimens (**b**).

**Figure 15 materials-14-02649-f015:**
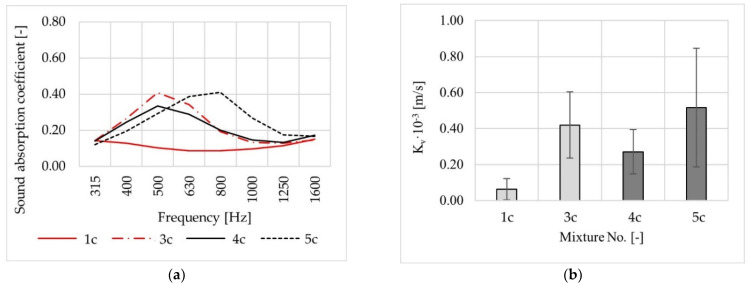
Test results for cylindrical specimens of mixtures chosen for placement on test sections: sound absorption coefficient (**a**), vertical water permeability (**b**).

**Figure 16 materials-14-02649-f016:**
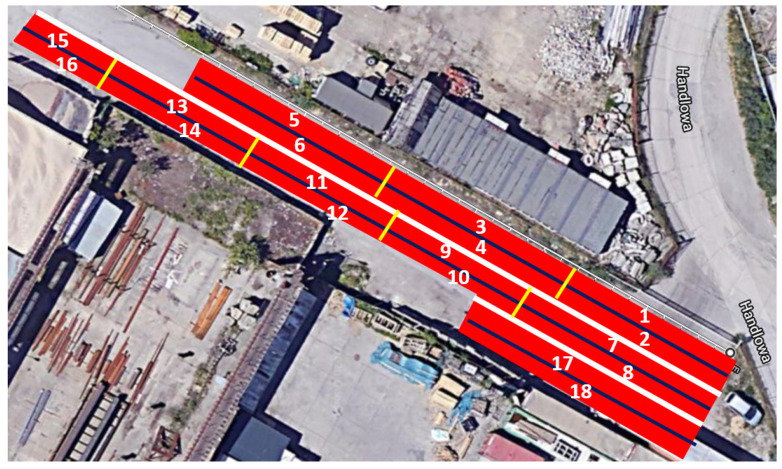
Layout of the test sections.

**Figure 17 materials-14-02649-f017:**
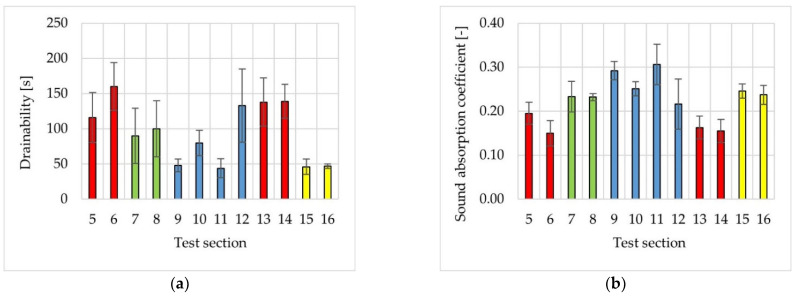
The results of water drainability (**a**) and sound absorption coefficient (**b**) on test sections.

**Figure 18 materials-14-02649-f018:**
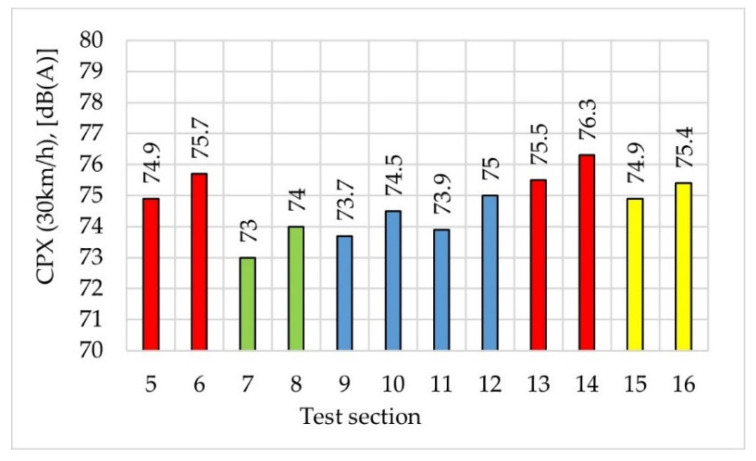
CPX index on test sections.

**Figure 19 materials-14-02649-f019:**
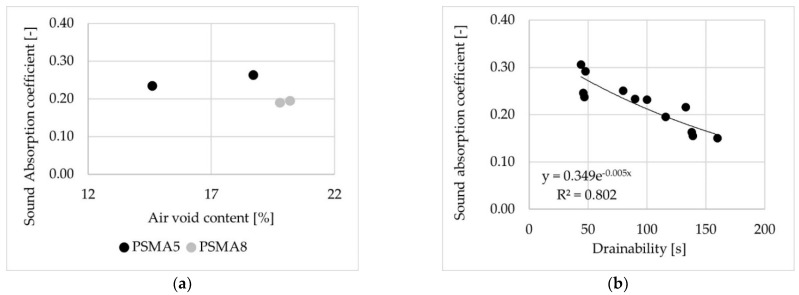
Sound absorption coefficient on the test sections, in relation to: (**a**) air void content, (**b**) water drainability.

**Table 1 materials-14-02649-t001:** Summary of different poroelastic mixtures selected for laboratory acoustic tests.

Mixture Designation	Maximum Grain Size [mm]	Type of Binder	Addition of Hydrated Lime	Crumb Rubber Gradation [mm]
1	8	45/80-80	No	2/4
2	8	45/80-80	Yes	2/4
3	8	45/80-80	No	4/5,6
4	5	45/80-80	No	2/4
5	5	45/80-80	Yes	2/4
6	5	65/105-80	No	2/4
7	5	65/105-80	Yes	2/4

**Table 2 materials-14-02649-t002:** Air void content and internal shear strength of the tested mixtures.

Mixture Designation	Air Voids Content (SD) [%]	Internal Shear Strength [MPa]
1	16.1 (± 1.8)	0.67
2	18.6 (± 0.5)	0.72
3	22.5 (± 2.1)	0.52
4	20.4 (± 0.8)	0.55
5	24.2 (± 1.8)	0.61
6	22.3 (± 0.8)	0.49
7	25.2 (± 0.8)	0.47

**Table 3 materials-14-02649-t003:** Combinations of mixture variants and construction factors.

Section No. (Mixture No.)	Paver Type	Screed Settings	Number of 8.5-ton Roller Passes
5 (No. 1)	small size	tampers only	3
6 (No. 1)			7
7 (No. 4)	standard size	tampers only	3
8 (No. 4)			7
9 (No. 5)		tampers only	3
10 (No. 5)			7
11 (No. 5)		tampers and vibrating plates	3
12 (No. 5)			7
13 (No. 1)		tampers only	3
14 (No. 1)			7
15 (No. 3)		tampers only	3
16 (No. 3)			7

## Data Availability

The data presented in this study are available in the databases of the authors at the Bialystok University of Technology and Gdansk University of Technology.

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
