# Peer review of "Investigation of Acoustic Properties of Poroelastic Asphalt Mixtures in Laboratory and Field Conditions"

_materials, 2021, doi:10.3390/ma14102649_

Round 1

Reviewer 1 Report

The topic is quite interesting. However, some minor revisions are required before accepted for publication. Please revise the manuscript as per the recommendations given. This paper requires thorough proofreading. Please check the entire paper for the grammatical errors.
1. Please include some of the key results in the abstract.
2. The introduction needs to elaborate. The references need to elaborate. It needs to add some references. Some suggestions reverences are below. Consider to add it. Thank you.
(1) Audrius Vaitkus, Ovidijus Šernas, Viktoras Vorobjovas, Judita GražulytÄ—. Selection of Constituent Materials for Asphalt Mixtures of Noise-Reducing Asphalt Pavements. The Baltic Journal of Road and Bridge Engineering 2019, 14(2),178-207. DOI: 10.7250/bjrbe.2019-14.439
(2) Wang, W.; Cheng, Y.; Chen, H.; Tan, G.; Lv, Z.; Bai, Y. Study on the Performances of Waste Crumb Rubber Modified Asphalt Mixture with Eco-Friendly Diatomite and Basalt Fiber. Sustainability 2019, 11, 5282. https://doi.org/10.3390/su11195282
(3) Copetti Callai, S.; Sangiorgi, C. A Review on Acoustic and Skid Resistance Solutions for Road Pavements. Infrastructures 2021, 6, 41. https://doi.org/10.3390/infrastructures6030041
3. The quality of the figures needs to be further improved.
4. Lack of discussion on the findings. Authors just reporting their findings.
5. The conclusions of the manuscript need to enhance a little.

Author Response

Reviewer 1

The topic is quite interesting. However, some minor revisions are required before accepted for publication. Please revise the manuscript as per the recommendations given. This paper requires thorough proofreading. Please check the entire paper for the grammatical errors.

Thank you for your remark. The article has been checked in terms of language correctness.

  1. Please include some of the key results in the abstract.

Additions have been made to the abstract.

  1. The introduction needs to elaborate. The references need to elaborate. It needs to add some references. Some suggestions reverences are below. Consider to add it. Thank you.
    (1) Audrius Vaitkus, Ovidijus Šernas, Viktoras Vorobjovas, Judita GražulytÄ—. Selection of Constituent Materials for Asphalt Mixtures of Noise-Reducing Asphalt Pavements. The Baltic Journal of Road and Bridge Engineering 2019, 14(2),178-207. DOI: 10.7250/bjrbe.2019-14.439
    (2) Wang, W.; Cheng, Y.; Chen, H.; Tan, G.; Lv, Z.; Bai, Y. Study on the Performances of Waste Crumb Rubber Modified Asphalt Mixture with Eco-Friendly Diatomite and Basalt Fiber. Sustainability 2019, 11, 5282. https://doi.org/10.3390/su11195282
    (3) Copetti Callai, S.; Sangiorgi, C. A Review on Acoustic and Skid Resistance Solutions for Road Pavements. Infrastructures 2021, 6, 41. https://doi.org/10.3390/infrastructures6030041

Thank you for your remark. The introduction has been completed and references to new literature have been added. The introduction discusses the main problems contained in the cited literature. The revised version of the manuscript shows these changes and added references.

  1. The quality of the figures needs to be further improved.

According to the authors, the figures were prepared in accordance with the requirements of the Editorial Board. We believe that they are legible and contain all the important information.

  1. Lack of discussion on the findings. Authors just reporting their findings.

Section 4 of the manuscript has been supplemented with a discussion of the results.

  1. The conclusions of the manuscript need to enhance a little.

Additions have been made to the conclusions.

Reviewer 2 Report

This is an interesting work. Some comments:

  1. Introduction can be improved, for example the benefits of using the noise reduction pavement can be rich.
  2. please include rubber in the title.
  3. lines 107 to 113: it is better to use a table here with the explanation of abbreviations.
  4. can the authors please provide the grading curves or cite the standards used?
  5. Figure 2: actual values is not visible in a figure, please use a table here.
  6. How about ITS and ITSR?
  7. Testing results and analysis are reliable.
  8. please provide future perspectives if possible.

Author Response

Reviewer 2:

This is an interesting work.

Thank you very much for the approval of the manuscript.

Some comments:

  1. Introduction can be improved, for example the benefits of using the noise reduction pavement can be rich.

Additions have been made to the introduction.

  1. Please include rubber in the title.

According to the authors, the title contains the phrase "... poroelastic asphalt mixtures ....", which is synonymous with inclusion of crumb rubber in the mixture. This is also covered in the Introduction. Therefore, the title of the article has not been changed.

  1. lines 107 to 113: it is better to use a table here with the explanation of abbreviations.

Thank you for your remark. A proper table with clearer explanation of the variants that were used has been added to the text. This table also replaces a part of the text previously found in lines 114-119.

  1. can the authors please provide the grading curves or cite the standards used?

Proper gradation curves have been added and presented in Fig. 1a, 1b and 1c.

  1. Figure 2: actual values is not visible in a figure, please use a table here.
  2. How about ITS and ITSR?

Explanation to Notes 5 and 6: A proper table has been added to the text. This table includes not only the values of air voids but also the values of internal shear strength (ITS) measured using Leutner test device. Indirect tensile strength was not used in evaluation of poroelastic mixtures, because it turned out that with such an elastic material this method was inadequate.

  1. Testing results and analysis are reliable.

Thanks for your comment.

  1. Please provide future perspectives if possible.

Additions have been made to the conclusions.

Reviewer 3 Report

In this paper, the authors investigated the acoustic properties of porous asphalt mixtures. The work is interesting and it is helpful for practitioners. I appreciated the empiric relationships between drainability, permeability, sound absorption and porosity. I think they could be useful for a qualitatively analysis.

Please find below some suggestions and comments that I hope you will take into consideration:

1) Line 45: other parameters that affects the acoustic efficiency are the tortuosity and the resistivity of the air. Here some papers that you can use as reference to introduce these parameters:

- S. Alber, W. Ressel, P. Liu, J. Hu, D. Wang, M. Oeser, D. Uribe, H. Steeb, Investigation of microstructure characteristics of porous asphalt with relevance to acoustic pavement performance, International Journal of Transportation Science and Technology, Volume 7, Issue 3, 2018, Pages 199-207, ISSN 2046-0430;

- Praticò, F.G., Vizzari, D., Fedele, R.: Estimating the resistivity and tortuosity of a road pavement using an inverse problem approach. 24th International Congress on Sound and Vibration, ICSV 2017;

- Attenborough, K., & Howorth, C. (1990). Models for the acoustic characteristics of porous road surfaces. In INTROC 90-International Tire/Road Noise Conference 1990, Gothenburg, Sweden.

2) Line 73: the word “many studies” is too generic, try to summarize in few lines the main content of these studies.

3) Figure 3: is the ACUPAVE system simply supported on the asphalt slab? Did you use any insulation material along the contact surface between the ACUPAVE system and the slab?

4) Line 155: based on my experience, the reliability of the measurements is influenced by the insulation of the lateral surface of the sample. How did you insulate your samples? Which material did you use?

5) Chapter 4: The difference of results between Kundt in lab and measurement in situ is not surprising. In general, the Kundt in lab has the best accuracy, even if it is time consuming.

6) In Figure 11, you calculated the average of sound absorption for the entire range of frequency. I suggest you to focus just on the range 400-630 Hz and 800-1250 Hz, which are the predominant one for tyre/road noise. Is it confirming the ranking that you observed?

7) In Figure 13, which sound aborption coefficient did you used? I suggest you to always refer to the Kundt in lab, because it is the best to detect the air void content. For more info about the reliability of this device, you can enrich your literature review with the following papers:

- Tan Li: Influencing Parameters on Tire–Pavement Interaction Noise: Review, Experiments and Design Considerations. Designs 2018, 2, 38; doi:10.3390/designs2040038;

- Filippo G. Praticò, Rosario Fedele, Domenico Vizzari, Significance and reliability of absorption spectra of quiet pavements, Construction and Building Materials, Volume 140, 2017, Pages 274-281

-Haitao Zhang, Zuoqiang Liu, Xiangchen Meng, Noise reduction characteristics of asphalt pavement based on indoor simulation tests, Construction and Building Materials, Volume 215, 2019, Pages 285-297

8) In conclusion, some lines about further researchers are missing.

Author Response

Reviewer 3

In this paper, the authors investigated the acoustic properties of porous asphalt mixtures. The work is interesting and it is helpful for practitioners. I appreciated the empiric relationships between drainability, permeability, sound absorption and porosity. I think they could be useful for a qualitatively analysis.

Thank you very much for the approval of the manuscript.

Please find below some suggestions and comments that I hope you will take into consideration:

1) Line 45: other parameters that affects the acoustic efficiency are the tortuosity and the resistivity of the air. Here some papers that you can use as reference to introduce these parameters:

- S. Alber, W. Ressel, P. Liu, J. Hu, D. Wang, M. Oeser, D. Uribe, H. Steeb, Investigation of microstructure characteristics of porous asphalt with relevance to acoustic pavement performance, International Journal of Transportation Science and Technology, Volume 7, Issue 3, 2018, Pages 199-207, ISSN 2046-0430;

- Praticò, F.G., Vizzari, D., Fedele, R.: Estimating the resistivity and tortuosity of a road pavement using an inverse problem approach. 24th International Congress on Sound and Vibration, ICSV 2017;

- Attenborough, K., & Howorth, C. (1990). Models for the acoustic characteristics of porous road surfaces. In INTROC 90-International Tire/Road Noise Conference 1990, Gothenburg, Sweden.

Thank you for your remark. Information about tortuosity and resistivity of the air has been added to the introduction. References to new literature have been added.

2) Line 73: the word “many studies” is too generic, try to summarize in few lines the main content of these studies.

The introduction discusses the main problems contained in the cited literature.

3) Figure 3: is the ACUPAVE system simply supported on the asphalt slab? Did you use any insulation material along the contact surface between the ACUPAVE system and the slab?

Yes, we used insulation material along the contact surface between the ACUPAVE system and the slab. To obtain a tight seal between the ACUPAVE tube and the pavement, the pavement attachment fixture and a soft material for insulation is used. For this purpose, we use plumbing putty, which is placed in a special groove on the bottom of the pavement attachment fixture.

4) Line 155: based on my experience, the reliability of the measurements is influenced by the insulation of the lateral surface of the sample. How did you insulate your samples? Which material did you use?

Yes, we used insulation with a soft foil roller and plumber putty of the edge and lateral surface of the sample during laboratory tests in the Spectronics ACUPAVE System adapter (Kundt’s tube). The ISO standard states that ‘sealing of any crack about the edge of the sample with vaseline or plasticine is recommended’, so we used this method. In the investigated poroelastic samples, this problem was particularly visible because the samples after drilling had a very jagged lateral surface.

5) Chapter 4: The difference of results between Kundt in lab and measurement in situ is not surprising. In general, the Kundt in lab has the best accuracy, even if it is time consuming.

Thank you for your remark. We are constantly working on finding and understand the relationship between the results obtained with the two methods (Kundt tube and measurement on the slab or in situ surface). We agree that the Kundt method has the best accuracy, however, conditions of slab tests are closer to the sound absorption conditions on real road surfaces.

6) In Figure 11, you calculated the average of sound absorption for the entire range of frequency. I suggest you to focus just on the range 400-630 Hz and 800-1250 Hz, which are the predominant one for tyre/road noise. Is it confirming the ranking that you observed?

Thank you for your remark. We agree that in the case of tyre/road noise the ranges of 400-630 Hz and 800-1250 Hz are the predominant. The ranking of the tested mixtures was compared for the entire range of frequency and for the range 400-630 Hz and 800-1250 Hz. The rankings were found to be almost the same for the 3 analyzed frequency ranges. Of course, the differences between the average values of tsound absorption coefficient are greater, in particular for the range 400-630 Hz for cylindrical samples. It is also visible in Figures 9 and 10. Taking into account the slight differences in the ranking of the mixtures, the mean values of sound absorption coefficient for the entire range of frequency were used in further analyzes.

7) In Figure 13, which sound aborption coefficient did you used? I suggest you to always refer to the Kundt in lab, because it is the best to detect the air void content. For more info about the reliability of this device, you can enrich your literature review with the following papers:

- Tan Li: Influencing Parameters on Tire–Pavement Interaction Noise: Review, Experiments and Design Considerations. Designs 2018, 2, 38; doi:10.3390/designs2040038;

- Filippo G. Praticò, Rosario Fedele, Domenico Vizzari, Significance and reliability of absorption spectra of quiet pavements, Construction and Building Materials, Volume 140, 2017, Pages 274-281

-Haitao Zhang, Zuoqiang Liu, Xiangchen Meng, Noise reduction characteristics of asphalt pavement based on indoor simulation tests, Construction and Building Materials, Volume 215, 2019, Pages 285-297

Thank you for your remark. Fig. 13 (renumbered Fig. 12) has been supplemented with the relationship between sound absorption coefficients determined on cylindrical samples and a comment has been added below the current Fig. 12. References to new literature have been added. The presentation of the results in chapter 4 has been slightly improved.

8) In conclusion, some lines about further researchers are missing.

Additions have been made to the conclusions.

Round 2

Reviewer 3 Report

Dear authors,

thanks for taking into account my suggestions. I think that the paper has been improved.

Regards